# Protein–Protein Interfaces: A Graph Neural Network Approach

**DOI:** 10.3390/ijms25115870

**Published:** 2024-05-28

**Authors:** Niccolò Pancino, Caterina Gallegati, Fiamma Romagnoli, Pietro Bongini, Monica Bianchini

**Affiliations:** Department of Information Engineering and Mathematics, University of Siena, Via Roma, 56, 53100 Siena, Italy; c.gallegati@student.unisi.it (C.G.); pietro.bongini@unisi.it (P.B.); monica@diism.unisi.it (M.B.)

**Keywords:** protein–protein interaction, bioinformatics, artificial intelligence, deep learning, graph neural networks, protein graph, protein interface

## Abstract

Protein–protein interactions (PPIs) are fundamental processes governing cellular functions, crucial for understanding biological systems at the molecular level. Compared to experimental methods for PPI prediction and site identification, computational deep learning approaches represent an affordable and efficient solution to tackle these problems. Since protein structure can be summarized as a graph, graph neural networks (GNNs) represent the ideal deep learning architecture for the task. In this work, PPI prediction is modeled as a node-focused binary classification task using a GNN to determine whether a generic residue is part of the interface. Biological data were obtained from the Protein Data Bank in Europe (PDBe), leveraging the Protein Interfaces, Surfaces, and Assemblies (PISA) service. To gain a deeper understanding of how proteins interact, the data obtained from PISA were assembled into three datasets: *Whole*, *Interface*, and *Chain*, consisting of data on the whole protein, couples of interacting chains, and single chains, respectively. These three datasets correspond to three different nuances of the problem: identifying interfaces between protein complexes, between chains of the same protein, and interface regions in general. The results indicate that GNNs are capable of solving each of the three tasks with very good performance levels.

## 1. Introduction

Proteins are large, biopolymeric molecules that are fundamental players in all living beings, with a variety of both structural and functional roles in cellular processes. Protein function is actually closely related to protein structure, which can be described on four different levels: the primary structure, namely the amino acidic sequence; the secondary structure, which is the local conformation of the peptide chain, consisting of helices, sheets, or coils; the tertiary structure, specifically the three-dimensional configuration of the molecule; and the quaternary structure, formed by two or more tertiary structure subunits that bind together, creating a protein complex. The smallest quaternary structure is a dimer, which is formed by two subunits, i.e., two monomers: if the two subunits have a similar nature, the protein complex is called a homodimer; otherwise, the formation of a heterodimer occurs. The amino acidic sequence determines the very final structure of the protein, which, in turn, dictates its biochemical functions. The analysis of the structural features of proteins is often helpful in understanding the biological processes that they are involved in, which ligands they bind to, and which molecular complexes they form [1].

Since proteins are responsible for various cellular functions, and many of these functions require the interaction of more protein molecules rather than individuals acting alone [2], protein–protein interaction (PPI) represents a key aspect of their activity. Signal transduction, i.e., a complex process involving PPIs that transmits information inside the cell to regulate gene expression [3], the binding between antibodies and protein antigens in the immune response [4], or the role of cellular adhesion proteins in maintaining tissue integrity [5] are only a few evident examples suggesting that a deeper knowledge about PPIs is critical. In addition, understanding these interactions is crucial not only for deciphering the complex orchestration of biological systems but also for the identification of novel therapeutic targets for disease intervention [6]. In addition, predicting which residues are involved in PPIs may help drug discovery, improve the accuracy of protein–protein docking, and obtain a richer annotation of protein function [7].

With the emerging developments in the machine learning and deep learning fields, an increasing number of researchers have explored their application to PPIs, since conventional experimental approaches for PPI prediction and site identification are expensive and time-consuming. Initially, computational methods mainly used common statistical features and highly conserved patterns in polypeptides since many functionally important proteins are preserved across species. [8]. The underlying idea is that proteins sharing homologous sequence patterns or structures may have a tendency of possessing the same interaction properties. The real consistent application of machine learning methods in PPI prediction can be traced back to 2001 [9]. In this phase, algorithms including decision trees [10], naive Bayes [11], random forests [12], and support vector machines [13] were employed. The following decade met a large spread of deep learning techniques applied to the PPI research field, showing far better results. Deep multi-layer perceptrons (MLPs) [14] and convolutional neural networks [15] were two of the most used methods in this phase. Subsequently, in more recent years, Yang et al. [16] were the first to apply graph convolutional networks (GCNs), i.e., a particular type of graph neural network (GNN), to the prediction of PPIs. This approach displayed particularly good performance since GNNs deal with graph-structured data with minimal information loss, and the complex nature of protein macro-molecules can be naturally represented with graphs. As a consequence, GNNs are shown to be a versatile and powerful tool for identifying specific interaction sites on proteins, or predicting the existence of interactions between protein pairs [6]. In addition, graph structures allow one to obtain protein representations from 3D Cartesian coordinate systems, maintaining spatial information and preserving multi-level rotation and translation equivariance. Therefore, graph neural networks, in all their various combinations and shapes [17,18,19,20,21,22], represent one of the most promising and performing architectures in such a wide realm of computational tools.

The methods for computational PPI site prediction can be roughly classified into two classes according to the information used for prediction [23]. The first class consists of sequence-based approaches that require readily available protein sequences. These generally have limited predictive accuracy due to the lack of structural and geometrical information [24]. The second class includes structure-based approaches that infer PPI sites from known structures. These are often more accurate and practically useful because proteins with unknown functions can be taken into account as well. These studies typically formulate the PPI problem as a graph-based learning task, such as node classification, link prediction, or graph classification, employing suitable GNN architectures to carry out the task. Nodes can represent amino acidic residues, i.e., what remains when two or more amino acids combine to form a peptide and the elements of water are removed, or secondary structures, depending on the problem at hand.

This study is proposed as the natural extension and evolution of [1]. Hence, a large amount of generic dimers, in both homogeneous and heterogeneous settings, have been retrieved from the PDBePISA database to build the graph-based dataset. Instead of considering a *correspondence graph* in the identification of biologically significant cliques, in this new perspective, protein complexes that directly represent the peptide chains of the original dimer are fed as input into GNNs. Nodes represent residues and the final task is the binary classification of interface residues. A structure-based approach is used, since dimers are represented as graphs in which residue nodes interact also with others located outside their immediately close neighborhood. In addition, three different implementations are exploited in order to involve the same dataset from different points of view and to try to understand which configuration is more useful to the GNN in the learning phase. The first considers all the dimer chains, with both self-interactions and interactions between different chains. The second takes into account pairs of interacting chains belonging to the same protein complex, with self- and reciprocal interactions. The last one involves single chains of the same dimer, excluding any type of interaction among different chains.

The rest of this paper is organized as follows: Section 2 shows the experimental results, which are contextually discussed. Then, Section 3 presents the materials and methods involved in the study, hence the GNN model, the dataset and its preprocessing phase, and the experimental setting. In addition, it presents an ablation study: a series of additional experiments carried out on subsets of the original features, both for nodes and edges, in order to evaluate the performance of the models in the absence of specific groups of attributes. Moreover, it provides a detailed example of a use case, in order to show the practical employment of a protein file, from the necessary processing phase to the corresponding output of the model. Finally, Section 4 draws some conclusions on current research and explains future perspectives.

## 2. Results and Discussion

PPI prediction is defined as a node-focused binary classification task using a GNN. The goal is to determine whether a generic node (residue) belongs to a protein interface or not. To test the predictive capabilities of the GNN model and evaluate its performance from different perspectives, experiments were carried out on the three different datasets. For each of these, a grid-search procedure was conducted to select the best model configuration. Performances of the best architecture for each dataset are reported in Table 1.

The best model configurations obtain very good results on all the datasets. All metrics show very high values for the *Interface* and *Whole* datasets, meaning that GNNs are capable of efficiently answering the two corresponding questions (see Section 3.2) on a per-residue basis. In particular, the best result is achieved in the *Whole* dataset, in which the model is capable of recognizing about 81% of interacting nodes while maintaining a high precision value of around 90% and resulting in the highest F-score and balanced accuracy values, scoring approximately 86% and 90%, respectively. However, promising results are still achieved in the *Interface* dataset, in which only pairs of interacting chains are considered: the key interpretation lies in the different number of nodes belonging to the two datasets, as the second one is composed of a greater number of nodes, resulting in a slight drop in performance. Not surprisingly, the more general question, arisen from the *Chain* dataset, is the most difficult to answer. In this case, the binary classification is based solely on the nature of the residue, the local structure of the protein, and the features of neighbor nodes and edges, without any conditioning on the chain or other protein, which should bind to the interface that the residue is part of. Nevertheless, even if the performance is not on par with the other two datasets, the values of all the metrics reach very good levels also for the *Chain* dataset, in which the model is able to recognize almost 68% of the interacting nodes, with an F-score of more than 60%.

## 3. Materials and Methods

In this section, the data and the experimental methodology used in this work are described. In particular, Section 3.1 presents an overview of the GNN model, the architecture used for PPI prediction. Section 3.2 proposes a precise description of the dataset obtained from the preprocessing phase (also described here) and the subsequent graph generation procedure. Subsequently, Section 3.3 shows the experimental setup chosen for the study, with specific reference to data setting, model selection, and evaluation. Section 3.4 presents an ablation study, which is a series of additional experiments to further investigate the influence of specific node/edge features from the dataset on the model’s predictive performances. Finally, Section 3.5 shows a practical use case, with a visual representation of the model predictions on a specific protein.

### 3.1. The GNN Model

A graph is a non-linear, non-Euclidean data structure defined as a tuple $G=(N,E,L_{N},L_{E})$, where *N* represents the set of nodes and E⊆N×N is the set of edges. Nodes describe entities, while edges stand for the relationships between them. Nodes and edges can be associated with values or vectors of values describing their attributes, and are defined as node feature vectors, $l_{n}\in L_{N},\;\forall n\in N$, and edge feature vectors, $e_{n,m}\in L_{E},\;\forall \left(n,m\right)\in E$, respectively.

In this context, GNNs represent a well-established paradigm within the field of machine learning, in which they have been successfully applied to an enormous variety of different tasks involving graph-structured data [25]. They exhibit a notable capacity for directly handling relational data in graph format while minimizing information loss and offering high flexibility [26]. Initial conceptualization of GNNs appeared in 2005 [27], followed by a comprehensive mathematical formulation in 2009, which also demonstrated that GNNs are universal approximators on graphs [28]. In the original work, GNNs operated by replicating the topology of the input graph and employing a message-passing mechanism between neighboring nodes to produce outputs on specific locations within the graph. This process was achieved by the means of two MLP units implementing as many parametric functions: a state transition function on each node of the graph and an output function where needed. In this context, problems can be categorized based on their focus on different elements within a graph: node-focused, edge-focused, or graph-focused. In node-focused problems, the model is asked to produce an output in correspondence with each node of the graph or a subset thereof, which can be used for classification, regression, or clustering purposes. Conversely, edge-focused problems concern tasks in which the output is produced either on each edge of the graph or on a subset of *E*. Finally, graph-focused tasks deal with problems in which a unique output is produced on the whole graph, and the goal is to predict a property or to cluster the complex object represented by the graph. In this latter case, the output is generally calculated on the nodes and then aggregated by sum or average operations to obtain a single output for each graph.

### 3.2. Dataset Description

The first step for building the dataset was to search for biological data in the Protein Data Bank in Europe (PDBe), leveraging the Protein Interfaces, Surfaces and Assemblies (PISA) service (all the information is available at https://www.ebi.ac.uk/pdbe/pisa/), an interactive tool provided by the PDBe database to study the stability of macro-molecular complexes (proteins, DNA/RNA, and ligands, accessed on 12 April 2024). This is essentially regulated by certain physicochemical properties, such as the free energy of the complex formation, hydrogen bonds, hydrophobic specificity, salt bridges across interfaces, solvation energy gain, and interface area. This information is used by the PISA service to analyze a given structure and to make predictions about a potential complex. It is possible to search for a protein using either a query, by indicating its PDB identification code, or to download entire databases. In this latest case, some parameters are available to filter the data, such as the type of interaction, keywords, presence of salt bridges or disulfide bonds, and symmetry number. To collect the dataset for the study, generic dimers characterized by a protein–protein interaction were retrieved from the PDBePISA database, for a total of 46,523 protein samples.

To guarantee biological significance, some criteria were enforced, such as a 〈x,y,z〉 symmetry, and only two interacting protein molecules at each interface. Subsequently, each sample was processed with the *Graphein* 1.7.0 Python package [29] to build the corresponding residue-based graph. Graphein is designed to streamline computational analysis for biomolecular structures, offering functionalities for constructing graph and surface mesh representations of various biomolecules, including proteins, nucleic acids, and small molecules. Additionally, Graphein facilitates the retrieval of structural data and chemical information from prominent bioinformatics databases such as the Protein Data Bank and ChEMBL. Therefore, a single graph was produced for each protein, with nodes representing the amino acid residues and edges representing the chemical bonds and forces between them, as depicted in Figure 1.

Nodes and edges are associated with feature vectors describing their attributes. In particular, residue-level features—each residue is represented by the corresponding carbon alpha atom $C_{\alpha }$—are collected into a vector $x\in R^{89}$ describing some physicochemical properties, such as their position vector in the three-dimensional space, the accessible surface area, the residue torsion angles, and DSSP [30], Meiler [31], and ExPASy [32] descriptors. A more detailed description is available at the link in the Appendix A. On the one hand, the Meiler representation [31] is a simplified model employed in computational biology and bioinformatics to depict the structural characteristics of amino acids. The method portrays each amino acid as a collection of geometric shapes approximating its backbone and side chain atoms, enabling the efficient analysis and modeling of protein structures and interactions in computational studies while reducing complexity. On the other hand, the ExPASy service [32], developed by the Swiss Institute of Bioinformatics (SIB), provides general non-residue specific features. It draws amino acid descriptors along the primary sequence of the protein, such as hydrophobicity, polarity, and molecular weight. DSSP [30] assigns each residue to a secondary structure element (SSE), such as α-helices, β-strands, and turns, by examining their hydrogen bonds and backbone geometries. This represents a standardized approach for characterizing protein secondary structures, with a total of 8 SSE classes, including the special class *random coil*, a conformation indicating the absence of a regular secondary structure. A generic edge between two nodes u,v∈N is represented by a feature vector $e_{u,v}\in R^{11}$, quantifying the distance between their $C_{\alpha }$ atoms (measured in angstrom Å) and describing the type of bonds established between the residues. These are represented as a non-categorical vector where a non-null value indicates the presence of a specific type of bond (e.g., multiple atom-level bonds between two residues are considered as parallel edges between residue-level $C_{\alpha }$ nodes, and are thus condensed in a single undirected edge). The ground truth for the classification task is obtained from PDBePISA, which defines a generic residue as interacting based on its solvation energy value. In particular, the solvation energy gain of the interface, expressed in kcal/M, is calculated as the difference in solvation energies of all residues between dissociated and associated (interfacing) structures. Therefore, a solvation energy equal to zero corresponds to non-interfacing residues, and a solvation energy different from zero corresponds to interfacing residues, i.e., located within the interfaces on the protein surface.

To test the predictive capabilities of the GNN model and gain some insight on chain interactions within proteins, three different alternative datasets were considered. The first dataset, which will be referred to as the *Whole (protein)* dataset, is composed of undirected graphs representing proteins in their entirety. All the interacting chains, possibly interconnected to let the information flow from a chain to another, are considered here, and are composed of 31,979 protein samples. Labels were associated to each node, representing the locations of all the interfaces within the protein at once. As shown in Figure 2, from a single protein graph belonging to this dataset, two other sets of objects can be extracted.

After slightly increasing the resolution of the dataset, in the second setting, each pair of interacting chains belonging to a generic protein was considered as a single sample. Each chain of the pair includes all its residues. In this context, repeated instances of the same chain can occur when this interacts simultaneously with two or more chains within the protein. Given that the interface location changes for each pair of interacting chains, the label associated to each node will be different. This dataset will be referred to as the *Interface* dataset, since a single graph represents the interaction between two single chains within the same protein, and it is composed of 73,427 interface graphs. In the last setting, further increasing the resolution of the graphs, each sample was represented by a single chain of residues, so, for a generic protein, only one instance of each chain could occur in the produced graphs. This dataset will be referred to as the *Chain* dataset and it is composed of 87,375 chain graphs, which include all their residues. More details about the datasets are reported in Table 2.

By treating a protein as a collection of coupling chains (i.e., using the *Interface* dataset), it is possible to focus the analysis on such critical interaction sites. This approach allows for a deeper understanding of how different protein components come together to form functional complexes. Identifying these interfaces is crucial for deciphering the molecular mechanisms underlying various cellular processes, including signal transduction, gene regulation, and immune responses. Conversely, the *Chain* dataset considers each chain within a protein independently, disregarding inter-chain interactions. While this simplifies the analysis, it provides insights into the individual behavior and properties of each chain within the protein ensemble. Eventually, considering proteins in their entirety as in the *Whole (protein)* dataset, without breaking them down into coupled or independent chains, is essential for understanding their overall structure, function, and behavior within biological systems. As a consequence, each dataset is investigated to answer a different biological question. The *Chain* dataset can provide an answer to the general question: which amino acids belong to interface regions in this peptide chain? This question is investigated by looking only at the characteristics of the amino acids and the structure of the peptide chain that they belong to. Thanks to the properties of GNNs, parts of the chain closer to the amino acid under analysis will be more influential in taking the decision with respect to other regions. The *Interface* dataset is instead ideal for answering the question: which amino acids play a role in the PPI between two peptide chains of the same protein, thus contributing to the formation of its own quaternary structure? In this scope, the binary classification of amino acids is conditioned to the particular PPI under analysis. The decision is based on the amino acid features and the structure of the chain, as well as on the specific structure of the interacting chain and the reciprocal position of the structures. Finally, the *Whole (protein)* dataset may answer the question: which amino acids play a role in PPIs between the protein complex that they belong to and other protein complexes? The decision is based on the features of the amino acid under analysis and the quaternary structure that it is a part of, additionally influenced by the characteristics of the other interacting structures and the relative positioning of the two protein complexes.

### 3.3. Experimental Setup

In this work, PPI prediction is modeled as a node-focused binary classification task using a GNN. The goal is to determine whether a generic node (residue) belongs to a protein interface or not. To test the predictive capabilities of the GNN model and evaluate its performance, experiments were carried out on the three different datasets: *Whole, Interface,* and *Chain*, described in Section 3.2. To maintain consistency across datasets, a seed was employed to split data into (non-overlapping) training, validation, and test sets in all the experiments in order to allocate the same proteins to the same sets in all three settings. As an example, the protein 3C3P shown in Figure 1 was assigned to the test set, and so were its extracted components, as described in Figure 2. Generally, in ML applications, the training set refers to the data used to train the model, and hence to adapt its parameters to the task at hand. The validation set is commonly used during the learning process for early stopping procedures, so as to avoid overfitting, i.e., when the model loses its generalization capabilities and specializes on the training data. Finally, the test set represents a never-before-seen set of data used for the evaluation of the model performance.

Usually, in a binary classification task, the performance of the model is measured in terms of accuracy; this metric, however, is not as precise on unbalanced datasets as the ones considered in this work. Hence, some other widely used metrics are also considered to evaluate the quality of the predictors, namely accuracy, balanced accuracy, recall, precision, F-score, and AUC (area under ROC curve). The different outcomes of predictions made by a generic model compared to the actual ground truth are described by the true positive, true negative, false positive, and false negative instances. On one hand, a true positive (TP) occurs when the model correctly identifies an instance belonging to the positive class, while a true negative (TN) ensues when it correctly identifies an instance belonging to the negative class. On the other hand, a false positive (FP) arises when the model mistakenly classifies a negative instance as positive, and a false negative (FN) occurs when it incorrectly labels a positive instance as negative. These distinctions are crucial for evaluating the accuracy and reliability of classification models, providing insights into their performance and potential areas for improvement. Therefore, the evaluation metrics correspond to the following:$$Accuracy=\frac{TP+TN}{TP+FP+TN+FN}$$
$$Balanced\;Accuracy=\frac{1}{2}\left(\frac{TP}{TP+FN}+\frac{TN}{TN+FP}\right)$$
$$Precision=\frac{TP}{TP+FP}$$
$$Recall=\frac{TP}{TP+FN}$$
$$F-\mathrm{Score}=\frac{2TP}{2TP+FP+FN}$$

Observe that the F-score can be interpreted as a weighted average of the precision and recall to provide a more balanced measure in cases of unbalanced datasets. In addition, balanced accuracy is defined as the average of the recall values obtained on each class. Finally, the AUC is the integral of the ROC curve and is equal to 1 for an ideal classifier and 0.5 for a random one, where the ROC curve corresponds to the plot of the true positive rate against the false positive rate at each threshold value. To find the best-performing architecture, an extensive grid-search procedure was carried out on the GNN hyperparameters; the search space as well as the best architecture for each setting in terms of F-score is reported in Table 3.

All the models were implemented with GNNKeras [33]—a Keras-based library for GNN recurrent architectures—and trained for 500 epochs with a cross-entropy loss function, an Adam optimizer with an initial learning rate of 0.001, and an early stopping procedure on the validation set. Moreover, since the ratio of negative/positive examples in the datasets is quite high, the multiple percentage of this ratio ($w_{bal}$) was considered in the grid search as a weight for positive examples to balance the learning process. Note that the training set is perfectly balanced with a value $w_{bal}=1$ (i.e., the total number of positive examples weighs as much as the total number of negative ones, so a single positive example is more important at training time than a negative one), while a value $w_{bal}=0$ means no balancing policy is used, so a positive example is as important as a generic negative example.

The best architectures in terms of F-score were further investigated in an ablation study to evaluate the impact on the model’s performance in identifying protein interaction sites by systematically removing a set of node/edge features from the datasets. This helps in determining which features are most informative for accurate predictions, guiding the refinement of predictive models for protein–protein interface identification. The procedure is described in detail in Section 3.4.

### 3.4. Ablation Study

To evaluate the importance of the contribution of the different feature components, an ablation study was carried out on the best architecture for each dataset setting. In particular, node features were grouped into four sets (3D position, Meiler, ExPASy, and DSSP components), while edge features were grouped into three sets (distance, weak chemical bonds, and strong chemical bonds). Experiments were conducted so as to eliminate one feature/edge group at a time from the datasets and evaluate the performance of the model in the absence of that group. The performance gap obtained by the new model gives an estimate of the importance of the features that were not considered during the learning process. The results of the ablation study on the three different datasets are reported in Table 4, Table 5 and Table 6.

Interestingly enough, the worst model performance is obtained when the DSSP component is excluded at training time, hence when no information about the secondary structure elements (SSEs) is provided to the models. In this context, although the recall score is comparable to that obtained in the previous study, missing information causes a general drop in performance over all the other metrics. Observe that the F-score is significantly affected, suffering from a drop of more than 14% in the worst scenario comparing to the original model. This outcome is in line with the expectations based on biological data, since the backbone torsion angles between adjacent amino acid residues in a protein chain are included among the SEE features. These angles determine the local conformation of the protein backbone, which affects its 3D structure and, consequently, the functional form of the protein. Variations in dihedral angles directly cause a change in the arrangement, affecting the folding pattern of the protein, which leads to the formation of the secondary structures. Additionally, the backbone torsion angles play a crucial role in determining the accessibility of amino acid residues to the surrounding solvent, hence impacting the protein’s accessible surface area and, as a result, its interfaces.

### 3.5. Use Case

A practical example of usage of the model is represented by the prediction of the interacting residues within the protein methyltransferase shown in Figure 1 and Figure 2, with PDB code 3C3P. In particular, the use case was performed on the *Whole* protein setting, as previously described in Section 3.2. The graph representation of the 3C3P protein was obtained using the Graphein Python package, described by 566 residue-based nodes, only 98 of which are located at the interfaces. The ground truth information for the interacting residues was retrieved from the PDBePISA database, which is used only to give an insight on the predictive capabilities of the trained model.

In inference mode, the trained model is able to successfully solve the task for each residue of the protein, with a total accuracy among the two classes of 97%, correctly classifying 550 residues and misclassifying a total of 16 residues within the protein. In particular, the model recognizes 90% of the interacting residues, with only nine interacting residues incorrectly classified as non-interacting.

Figure 3 shows a graphical representation of the predictions inferred by the model on the protein methyltransferase.

## 4. Conclusions

In this work, the biological problem of identifying the interfaces on the protein surface is modeled as a node-focused binary classification by means of a GNN, which is asked to determine whether a generic residue belongs to a protein interface or not. Model performance was measured in terms of F-score and balanced accuracy in three different settings and unbalanced dataset scenarios. The results suggest that the approach presented in this work is very promising, being able to score almost 86% and 90%, respectively, with, however, room for improvement. Moreover, an ablation study was carried out on the best architecture for each dataset, grouping nodes’ and edges’ features and confirming the expectations based on the biological importance of the descriptors.

Future research may possibly investigate the PPI prediction task on the three datasets from a different perspective: in the new context, the protein graph could be composed of nodes representing SSEs, such as α-helices and β-sheets, with sequence-specific features extracted by a sequence-based architecture, namely an LSTM model. Emerging GNN-based models have the potential to enhance the precision and scope of research in this field, and incorporating sequence-based models for residue-based sequences, together with GNNs for the protein structure, could open new avenues for investigating PPI prediction.

## Figures and Tables

**Figure 1 ijms-25-05870-f001:**
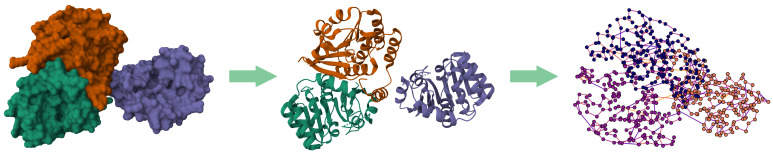
Example of a graph-based representation of the protein methyltransferase with PDB code 3C3P. The resulting graph on the right is composed of nodes representing the residues and edges representing the bonds between them. In the example, nodes are depicted with a gradient color scheme based on the afferent chain along the protein structure, from blue to orange, while edge color is defined by the number of interactions between residues. Nodes and edges are associated with feature vectors describing their properties in both residues and secondary structure granularity.

**Figure 2 ijms-25-05870-f002:**
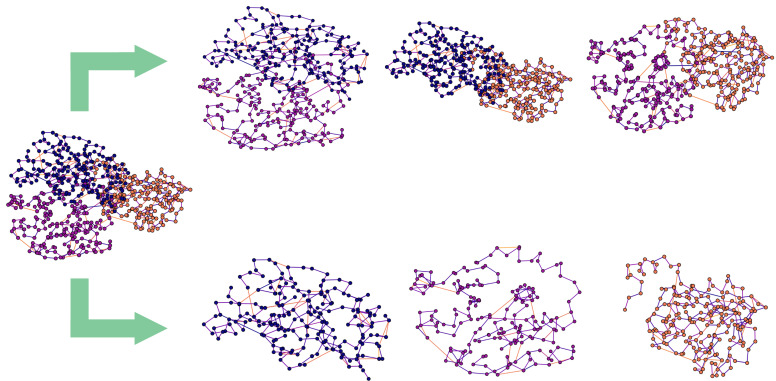
Example of sample extraction for different dataset settings. On the left, the single graph built on the protein with PDB code 3C3P as depicted in Figure 1, and belonging to the *Whole (protein)* dataset, in which a generic graph represents the entire protein; above, the extracted samples belonging to the *Interface* dataset, in which a single graph represents a couple of interacting chains; below, the extracted samples belonging to the *Chain* dataset, in which a single graph represents a single chain. Note that the *Interface* dataset includes more samples than the other two, since multiple instances of the same generic chain can occur, meaning that it can interact simultaneously with different chains: in this case, the labels are adapted to the specific interface instance. As a result, the label in the *Whole* and *Chain* datasets are obtained by the superposition of all chain instances among the interfaces.

**Figure 3 ijms-25-05870-f003:**
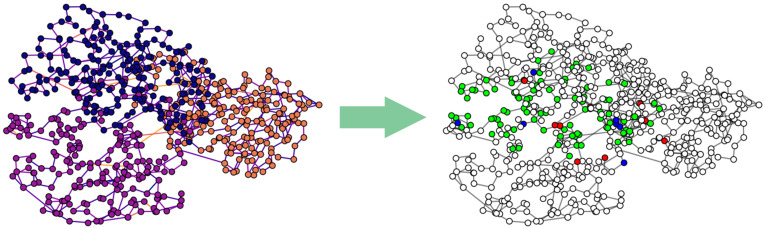
Example of a use case on the protein methyltransferase with PDB code 3C3P. On the left, the residue-based graph representation obtained with Graphein Python package, colored as stated in Figure 1; on the right, the output of the GNN model in the binary node-focused classification task, in which the model is asked to predict whether or not a residue is located in the interface. True positive (TP), true negative (TN), false positive (FP), and false negative (FN) predictions are colored in green, white, blue, and red, respectively. The model correctly classifies 461 non-interacting residues (TN) and 89 interacting residues (TP), while it fails to recognize 9 non-interacting residues (classified as interacting, i.e., FP) and 7 interacting residues (classified as non-interacting, i.e., FN).

**Table 1 ijms-25-05870-t001:** Performance of the best architecture for each dataset as a result of the grid-search procedure.

Dataset	Acc	BAcc	Precision	Recall	FScore	AUC
Whole	0.9467	0.8946	0.8982	0.8108	0.8522	0.9794
Interface	0.9610	0.8880	0.8627	0.7927	0.8262	0.9793
Chain	0.8335	0.7717	0.5454	0.6731	0.6025	0.8679

**Table 2 ijms-25-05870-t002:** Description of the datasets *Whole, Interface,* and *Chain*. *N* and *E* refer to the number of nodes and edges belonging to a single sample, respectively, while % res_*i*_ represents the percentage of interacting residues. Note that the total number of nodes in the *Interface* dataset is greater than the number of nodes in the other datasets, since multiple instances of the same chain can occur.

Dataset	Samples	$N_{\min}$	$N_{\max}$	$E_{\min}$	$E_{\max}$	% res_*i*_
Whole	31,979	10	9288	19	40,369	18.87
Interface	73,426	4	7197	2	28,567	11.83
Chain	87,375	2	3607	1	14,376	18.87

**Table 3 ijms-25-05870-t003:** Hyperparameter search space and best configurations obtained for the GNN model in the grid-search procedure, in all the dataset settings. Note that $w_{bal}$ refers to the fraction of the balancing weight that has been assigned to the minority class at training time; a value of $w_{bal}=0$ means that no balancing policy has been used at training time, while a value of $w_{bal}=1$ means that the classes have been perfectly balanced. Moreover, *h* refers to hidden layer units in the network state and output, respectively, net_*s*_ and net_*o*_, with a value of h=0 meaning that no hidden layers are used in the architecture.

Hyperparameter	Grid-Search	Best Configuration		
		Whole	Interface	Chain
state dimension	16, 32, 64	32	16	16
max iteration	3, 4, 5	5	5	5
$w_{bal}$	0, 0.2, 0.5, 1	0.5	0.5	1
net_*s*_ units *h*	0, 256, 512	256	256	256
net_*o*_ units *h*	0, 256	0	256	256

**Table 4 ijms-25-05870-t004:** Ablation study on the *Whole* dataset with the best architecture obtained from the grid-search procedure. The minimum value for each metric is highlighted in bold. Note that “None” refers to Table 1, in which all features are included in the learning process.

Excluded Group	Acc	BAcc	Precision	Recall	FScore	AUC
None	0.9467	0.8946	0.8982	0.8108	0.8522	0.9794
3D Position	0.9469	0.8953	0.8985	0.8120	0.8531	0.9793
Meiler	0.9432	0.8913	0.8824	0.8079	0.8435	0.9761
ExPASy	0.9434	0.8953	0.8753	0.8178	0.8456	0.9770
DSSP	**0.9041**	**0.8418**	**0.7501**	**0.7415**	**0.7458**	**0.9411**
distance	0.9446	0.8928	0.8882	0.8095	0.8470	0.9779
weak bonds	0.9450	0.8918	0.8935	0.8062	0.8476	0.9773
strong bonds	0.9392	0.8736	0.8966	0.7678	0.8273	0.9729

**Table 5 ijms-25-05870-t005:** Ablation study on the *Interface* dataset with the best architecture obtained from the grid-search procedure. The minimum value for each metric is highlighted in bold. Note that “None” refers to Table 1, in which all features are included in the learning process.

Excluded Group	Acc	BAcc	Precision	Recall	F-Score	AUC
None	0.9610	0.8880	0.8627	0.7927	0.8262	0.9794
3D Position	0.9582	0.8826	0.8476	0.7838	0.8145	0.9758
Meiler	0.9614	0.8890	0.8647	0.7944	0.8281	0.9791
ExPASy	0.9584	0.8861	0.8431	0.7918	0.8166	0.9776
DSSP	**0.9188**	**0.8464**	**0.6277**	**0.7518**	**0.6842**	**0.9327**
distance	0.9586	0.9036	0.8175	0.8317	0.8245	0.9793
weak bonds	0.9596	0.8879	0.8505	0.7942	0.8214	0.9771
strong bonds	0.9555	0.8796	0.8291	0.7805	0.8041	0.9715

**Table 6 ijms-25-05870-t006:** Ablation study on the *Chain* dataset with the best architecture obtained from the grid-search procedure. The minimum value for each metric is highlighted in bold. Note that “None” refers to Table 1, in which all features are included in the learning process.

Excluded Group	Acc	BAcc	Precision	Recall	F-Score	AUC
None	0.8335	0.7717	0.5454	0.6731	0.6025	0.8679
3D Position	0.8188	0.7577	0.5130	0.6598	0.5772	0.8526
Meiler	0.8324	0.7627	0.5443	0.6513	0.5930	0.8618
ExPASy	0.8139	0.7638	0.5028	0.6835	0.5794	0.8547
DSSP	**0.7528**	**0.7232**	**0.4048**	0.6758	**0.5063**	**0.8029**
distance	0.8492	0.7404	0.6045	**0.5662**	0.5847	0.8588
weak bonds	0.8480	0.7522	0.5940	0.5989	0.5964	0.8658
strong bonds	0.8248	0.7635	0.5260	0.6653	0.5875	0.8589

## Data Availability

The raw data supporting the conclusions of this article will be made available by the authors on request.

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
