# Peer review of "Protein–Protein Interfaces: A Graph Neural Network Approach"

_ijms, 2024, doi:10.3390/ijms25115870_

Round 1
Reviewer 1 Report
Comments and Suggestions for Authors
The authors suggest a GNN method to detect protein protein interfaces. Neural networks are a well established tool in bioinformatics, and the detection of protein-protein interface can be very helpful in docking, protein protein interactions and network studies.
The paper is well written and well motivated, but very descriptive. The technical side is lacking or not very clearly explained. Specific comments include:
1. How do you define interacting residues? Most methods put the threshold at 6 or 8 Angstroms.
2. There have been several methods for detecting PPI using neural networks, deep learning etc. The authors made no comparison to existing methods.
3. Some details are not well explained and should either be added, or alternatively be accompanied by a proper citation. For example: what constitutes the 89 dimension vector in section 2.2 (page 4, line 167) and the 11 feature vector on the same page, line 183?
4. Is the interface dataset defined as the entire two interacting chains or only the residues that are close by?
5. Minor comment: Figure 1's caption says that nodes are depicted in color gradient but it was very hard to see it in the figure.
Comments on the Quality of English LanguageThe English is overall good, but some sentences are long and required to read carefully several times.
Reviewer 2 Report
Comments and Suggestions for Authors
I found this to be an overall well-conceived, well-executed, and well-written work. I do have one broad suggestion that I feel could significantly improve the work: please provide a practical example of a use case, making it obvious what the user would need to input into the model, and what the output of the model would be. For example, choose a specific protein from the PDB, describe how that protein file needs to be prepared for input, and then demonstrate and explain what the model provides as output for that particular input protein.
Additional comments/questions:
Line 243: “training, test, and validation sets” Please define what each of these sets is, that is, what purpose does each of these sets serve?
